



# 1 Characteristics of Ground Ozone Concentration over Beijing from 2004 to 2015: Trends, Transport, and Effects of Reductions

Nianliang Cheng[1,2,3], Yunting Li [1],Dawei Zhang [1,4*],Tian Chen[5], Feng Sun[1,6],Chen Chen [1,6] ,Fan Meng[2,3]
[1]Beijing Key Laboratory of Airborne Particulate Matter Monitoring Technology,Beijing Municipal Environmental
Monitoring Center,Beijing 100048, China
[2]College of Water Sciences,Beijing Normal University,Beijing 100875,China
[3]State Key Laboratory of Environmental Criteria and Risk Assessment, Chinese Research Academy of Environmental
Sciences, Beijing 100012, China
[4]Department of Environmental Science and Engineering,Tsinghua University, Beijing 100084, China
[5]Beijing Municipal Environmental Protection Bureau, Beijing 100044, China
[6]Department of Environmental Science, Peking University, Beijing 100871, China
[*]Corresponding author at:Beijing Municipal Environmental Monitoring Center,Beijing 100048, China.E-mail address:
zhangdawei@bjmemc.com.cn (W.D. Zhang).
**Abstract:** Based on the hourly ozone monitoring data during 2004–2015 in urban area and at
DL background station in Beijing, a comprehensive discussion of the characteristics of ozone
concentration was conducted. Annual concentration of daily maximum 1 h ozone ($O_3$ 1h) was
all increasing at urban sites(1.79 ppbv yr$^{-1}$) and DL background station(2.05 ppbv yr$^{-1}$) while
daily maximum 8 h average ozone concentration($O_3$ 8h) was increasing in urban area(1.14
ppbv yr$^{-1}$) and slightly decreasing at DL background station(-0.47 ppbv yr$^{-1}$) from 2004 to
2015 due to different ozone sensitivity regimes and ratios of $NO_2$/NO.Diurnal variation of
ozone peaks obtained at downwind DL station were about 1 h later than that in urban area
from May to October in different years and concentration of ozone at a DL background
station was much higher than that of urban sites. Moreover, the difference of ozone peaks
between urban sites and DL background station was significantly becoming smaller in recent
years, which may be related to the regional ozone transport and the expansion urbanization of
Beijing. Based on the joint efforts of regional air pollution prevention and control,Beijing
achieved Sep 3 military blue.Calculated average concentrations of CO, $NO_2$, and $O_3$ in
S2(Aug 20~31,2015) and S3(Sep 01~03,2015) decreased by 31.48%, 43.97%, and 13.21% at
urban sites, and by 20.93%, 57.10%, and 23.62% at DL station, respectively compared with
those in S1(Aug 01~19,2015) and S4(Sep 04~30,2015).A reduction of local anthropogenic
emissions such as VOCs and $NO_x$ could reduce ozone efficiently especially in downwind
areas of Beijing and made the ozone peaks decrease significantly and appear 2~3h earlier
compared to the scenarios of no emission reductions. Compared to the increasing ozone
during Asia-Pacific Economic Cooperation (APEC)meeting period,to decrease the ozone
concentration in Beijing, VOCs emissions should be reduced larger and be controlled stricter





than that of $NO_x$ in Beijing and the policy of regional air pollution joint prevention and
control should still be promoted unswervingly and jointly in the further.

**Key words:** $O_3$8h; trend; Beijing; regional transport; reductions




## 1. Introduction

Ground-level ozone, one of the most important secondary air pollutants in the atmosphere, is generated through photochemical reactions between nitrogen oxides($NO_x$) and volatile organic compounds (VOCs) (Trainer et al., 2000; Sillman, 1999). High concentrations of ozone near the ground are harmful to human health, ecosystems, and global climate (Fiore et al., 2009).

In recent years, elevated regional ozone concentration and atmospheric oxidation capacity in China have attracted increasing attention (Lin et al., 2008; Zhang et al., 2007). Numerous studies have analyzed the concentration variations of ozone and its photochemical reactions with its precursors based on the measurements over a short period or satellite data (Chan et al., 2003; Wang et al., 2012; Vingarzan, 2004). Most studies in China were mainly concentrated in city cluster regions, such as Pearl River delta (Li et al., 2011;Wei et al., 2012; Zhang et al., 2013), Yangtze River delta (Li et al., 2014; Ding et al., 2013; Ran et al., 2009), and Beijing–Tianjin–Hebei regions(BTH) (Tang et al., 2009;Shao et al., 2009; Lu et al., 2010). These studies focused on the chemical characteristics of ozone, with few discussions on the variation trends and ozone transport and its influencing factors especially by regional reduction measures within a long period because of the lack of observed data in Beijing (An et al., 2006; Chou et al., 2009; Yuan et al., 2009) and other limiting facors.

Different to the continuously decreasing ground ozone concentrations in urban sites in the US (Pollack et al., 2013), recent limited studies performed in China, particularly in BTH area, suggested that ozone concentrations in both regional background and urban areas are increasing (Meng et al., 2009; Wang et al., 2008) due to large $NO_x$ emissions. Few long-term studies analyzed the trends of ground-level ozone in Beijing (Lu et al., 2010),let alone analyze the trends of daily maximum 8 h average ozone concentration($O_3$ 8h) and daily maximum 1 h ozone concentration($O_3$ 1h) and effects of urbanization and regional emission reduction measures on ozone concentrations. After the implementation of the new standard of "Ambient Air Quality Standard" (MEP, 2013) in 2013, the levels of $O_3$ 1h and $O_3$ 8h have a direct impact to the ranks of the air quality in Beijing. Furthermore, the increasing ozone pollution of Beijing obtained much public concerns from Beijing Municipal Government and the whole society (Ding et al., 2013; Wang et al., 2013) especially in Summer. The executive meeting of the State Council examined and adopted 'The Control Measures of Beijing Air Pollution during 2012–2020" (http://zhengwu.beijing.gov.cn/gzdt/gggs/t1225355.htm). According to the regulation, the non-attainment hours of ozone in Beijing will decrease by 30% than that in



2010 and should be controlled at about 200 hours annually. Therefore, the results of previous
studies were far from the current needs.
Air quality security programs were implemented from Aug 20 to Sep 3 in 2015 to
guarantee the air quality for the parade on the 70th Victory Memorial Day for the Chinese
People's War of Resistance against Japanese Aggression. Chinese government established
numerous emission reduction measures, such as reducing coals, industrial adjustment, joint
prevention measures, and limitation of vehicles (particularly heavy-duty buses and trucks
from outside Beijing, and odd-even license plate policy on roads within urban Beijing). As
regional emission reduction measures can not be copied and costs a lot of manpower and
material resources,it offers a precious opportunity to study the changes in ozone and its
precursors during the period of air quality assurance.
This paper aims to investigate the temporal trends of $O_3 1h$ and $O_3 8h$ in different sites in
Beijing and verify the importance of ozone transport. Also, we evaluated the changes on
ozone concentration after the reduction measures during the Sep 3 military parade in 2015.

**2 Materials and methods**
**2.1 Site distribution**
Beijing is located at 115.7 °–117.4 °E, 39.4 °–41.6 °N. This area is at the northwest edge of
the North China Plain and close to the edge of the semi-desert zone. Its terrain exhibits a
dustpan shape, and it is surrounded by mountains in three directions. The average altitude of
Beijing is 43.5 m, and the general altitude of mountains is in the range of 1 000–1 500 m,
which is not conducive to pollutant diffusion. The total area of Beijing is 16410.54 $km^2$, in
which 62% are mountains. Its total forest coverage in the plain region is about 15%, which is
lower than that in whole city (38%). Beijing exhibits a temperate continental monsoon climate,
where it is hot and rainy in summer and cold and dry in winter. Over the past decade, the
annual average rainfall is less than 450 mm, 80% of which is concentrated in June, July, and
August (BJEPB 2014; Beijing Statistics Bureau, 2014).
As the capital of China, the air quality monitoring network in Beijing is more advanced
than that of the remaining regions of China(BJEPB, 2014). In 2001, an air quality monitoring
network that obtains 35 monitoring stations was established by the Beijing Municipal
Environmental Monitoring Center (BJMEMC, http://zx.bjmemc.com.cn/, **Fig. 1**). The 35
monitoring stations cover all districts that contain different environment types defined by
regional background, such as suburbs, city, and residential. Twelve monitoring sites (DL, DS,
GY, TT, WSXG, AT, NZG, WL, GC, SY, CP, HR) in urban area and one background station





DL were selected in Beijing and used in this study. DL station (116.22 °E, 40.29 °N, about 45
km northwest of Tiananmen square) is the background station of World Meteorological
Organization Environmental Monitoring center in China and has conducted air pollutant
monitoring work for decades.Meteorological sounding data in Beijing at Guanxiangtai
station(GXT,54511)were downloaded from the Department of Atmospheric Science, College
of Engineering, University of Wyoming (http://weather.uwyo.edu/upperair/sounding.html).

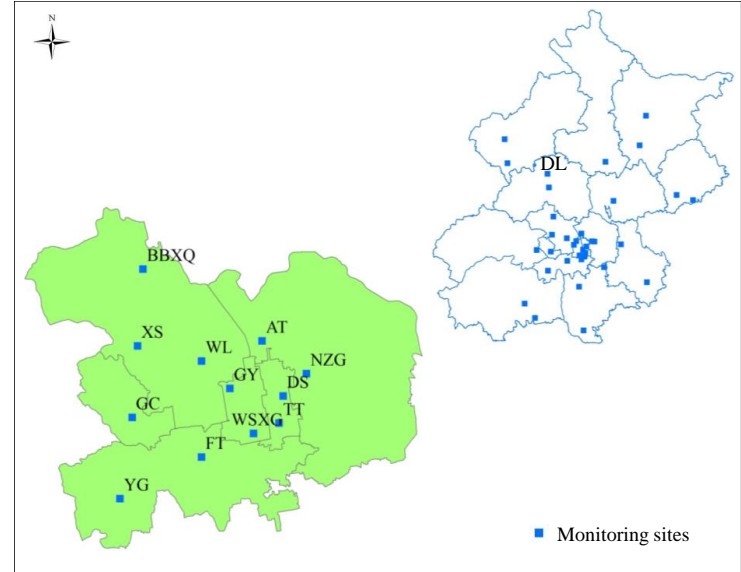


**Fig. 1** Distribution and classification of observation sites in Beijing

**2.2 Monitoring instruments**

The monitoring instruments of ozone are all the 49C ozone analyzer instruments

produced by Thermo  Fisher  Corporation (USA). The minimum limit of ozone analyzer
instrument is $1 \times 10^{-9}$, and the zero and cross drifts are 0.4%/24 h and ±1% /24 h, respectively.
An ozone calibrator (49IPS) traceable to the Standard Reference Photometer maintained by
the WMO World Calibration Center was used to calibrate the ozone analyzers. Ozone
monitoring instrument at each station had a zero cross calibration every three days, precision
audit every three month, and an accuracy check every six months to ensure the monitoring
quality of ozone in Beijing. Thermo Fisher 42C $NO–NO_2–NO_x$ analyzer was used to monitor
NO and $NO_2$ concentrations with a limit of $0.05 \times 10^{-9}$, zero drift of $0.025 \times 10^{-9}$/24 h, and span
drift of ±1%/24h. Operation procedure strictly followed the "The Specification of
Environmental  Air  Quality  Automatic  Monitoring  Technology"  (HJ/T193-2005,





http://kjs.mep.gov.cn/hjbhbz/bzwb/dqhjbh/jcgfffbz/200601/t20060101_71675.htm),  and  the
equipment was regularly calibrated and maintained by technicians.

**3 Results and Discussion**
**3.1 Variations trends**

A simple linear regression and statistical tests, such as Pearson's correlation analysis were

implemented to investigate the trends of $O_3$1h and $O_3$8h in urban area and at DL station in
Beijing (**Fig. 2**), for the interannual variations of ozone concentrations in urban Beijing, $O_3$1h
was in an evident upward trend with an annual concentration growth rate (AAGR) of 1.79
ppbv $yr^{-1}$ (correlation coefficient R=0.82,highly correlated) and an higher increase of 2.84
ppbv $yr^{-1}$ during May to September(MSAGR,R=0.87,highly correlated) from 2004 to
2015.Variation of $O_3$8h was in an overall upward trend with AAGR of 1.14 ppbv $yr^{-1}$
(R=0.88,highly correlated) and MSAGR of 1.68 ppbv $yr^{-1}$ (R=0.85,highly correlated) during
May to September, respectively from 2004 to 2015. For the variations of ozone concentration
at DL background station, $O_3$1h was in an overall upward trend with AAGR of 2.05
ppbv $yr^{-1}$(R=0.81, highly correlated) and MSAGR of 0.14 ppbv $yr^{-1}$(R=0.10, micro relevant),
whereas $O_3$8h was in a slightly downward trend (AAGR=−0.47 ppbv $yr^{-1}$, R=−0.42,weak
correlation,real relevant; MSAGR=−0.70 ppbv $yr^{-1}$, R=−0.40, real relevant).

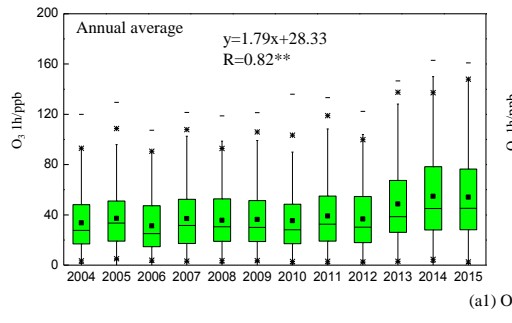

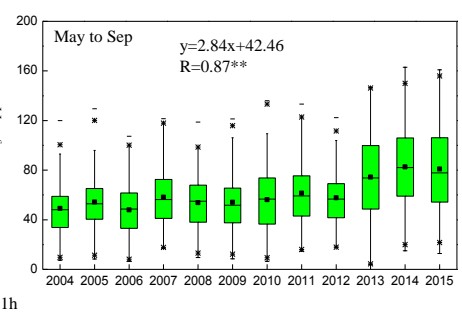

(a1) $O_3$1h

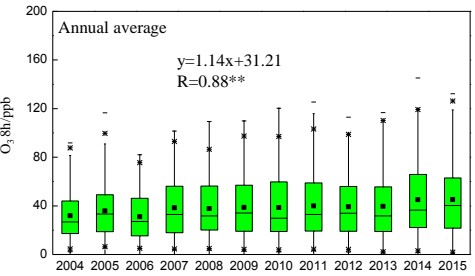

                                                                                                                (a2) $O_3$8h

(a) Urban



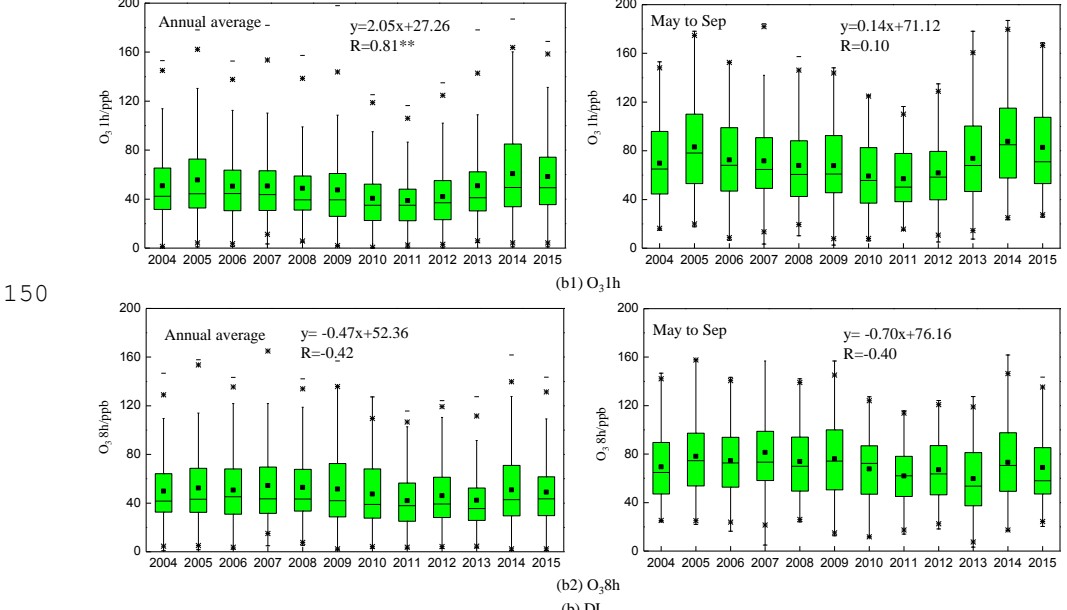

(b1) O$_3$1h

(b2) O$_3$8h

(b) DL

**Fig. 2** Variation trends of O$_3$1h and O$_3$8h in urban Beijing (a1, a2) and at DL background station (b1, b2) from 2004 to 2015
(including annual trends and trends from May to Sep) and the linear fitting equations(95% confidence interval, two-
tailed ,**highly relevant)

$$r = \frac{\sum (X - \overline{X})(Y - \overline{Y})}{\sqrt{\sum (X - \overline{X})^2 \sum (Y - \overline{Y})^2}} \quad (R1)$$

(r is the pearson correlation coefficient; X is the ozone concentration;Y presents the year;r between 0 and 0.3 representing the
micro relevant, r between 0.3 and 0.5 representing the real relevant, r between 0.5 and 0.8 representing the significant
relevant,and r between 0.8 and 1.0 representing highly relevant)

For the variations of O$_3$1h and O$_3$8h concentration in different periods at twelve sites in
urban Beijing and DL background site(**Fig.3**), concentrations of O$_3$1h and O$_3$8h were all
significantly increased during the period of 2013–2015 compared to  those during the periods
of 2004–2007 and 2008–2012. Average concentration of O$_3$1h during the period of 2013–
2015 increased 3.71%~40.29% at urban sites in Beijing compared to that during the period of
2004~2012 while average concentration of O$_3$8h during the period of 2013–2015 increased
9.51%~62.58% at urban sites in Beijing compared to that during the period of
2004~2012.Average concentration of O$_3$1h during the period of 2013–2015 exceeded  the
standard about 77.53%~104.55% at urban sites in Beijing while it was 33.09%~92.32% for
O$_3$8h.Therefore,the rising ozone concentration after the implementation of the new standards
highlighted the terrible ozone pollution situation in recent years in Beijing.





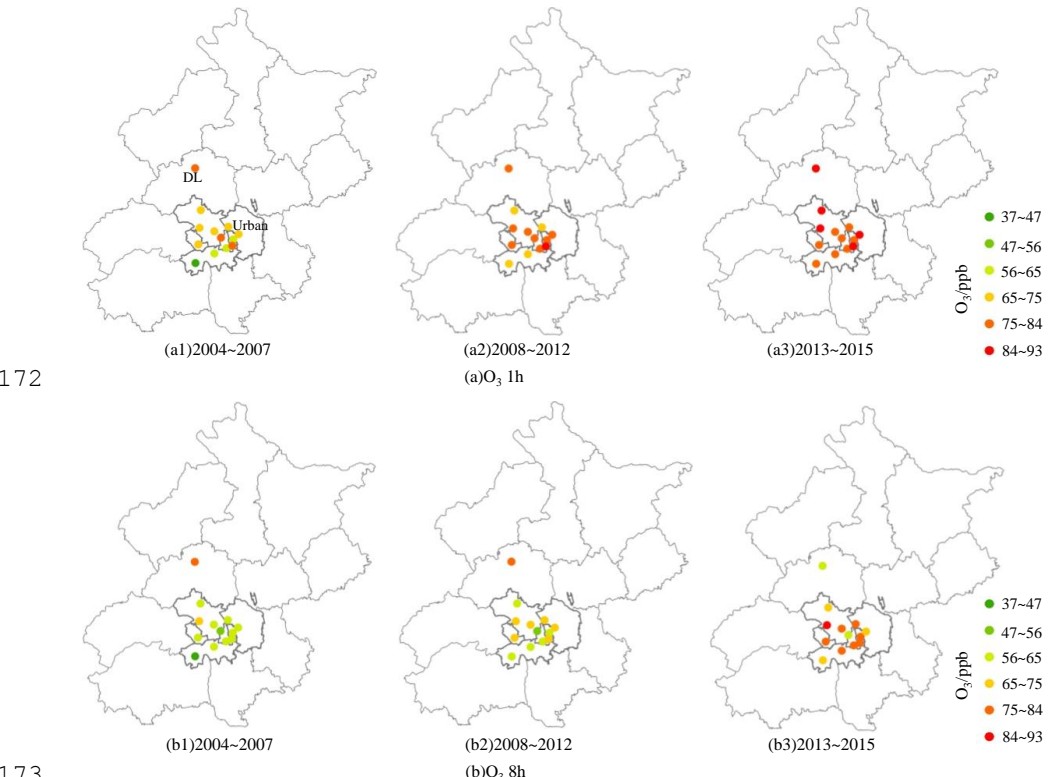



**Fig. 3** Average concentrations of $O_3$1h and $O_3$8h at urban sites and DL background site from May to Sep during the periods

of 2004–2007, 2008–2012, and 2013–2015 in Beijing


From the frequency distributions(proportion rates in different ozone concentration
intervals) of $O_3$1 h and $O_3$ 8h in Beijing(**Fig. 4**), frequency of $O_3$ 8h less than 10 ppbv
increased significantly during the period of 2013 to 2015 compared with that during period of
2008–2012 and frequency of ozone concentration higher than 80 ppbv also became larger
both at DL background station and urban sites.For $O_3$ 1h,frequency of $O_3$ 1h higher than 80
ppbv became significantly larger both at DL background station and urban sites during the
period of 2013 to 2015 compared with that during period of 2008–2012 which indicated
increasing frequencies of high ozone concentration caused the significant increase of $O_3$1h in
Beijing in recent years; whereas the low values of $O_3$1h did not increased as $O_3$8h. This
phenomenon was basically consistent with a few recent studies (Tang et al., 2009; Jonson et
al., 2006; Xu et al., 2008) and could explain the increasing high ozone concentrations in
Beijing due to the increasing frequency of higher ozone concentration at both background
station and urban sites. Parrish et al. (2014) found that ozone volume fraction increases at an
annual growth of 0.2–0.3 ppbv $yr^{-1}$ in the Northern Hemisphere. Also, Meng et al. (2009)





observed that ozone volume fraction increases at a rate of 1.0 ppbv $yr^{-1}$ at the background
station in Shanghai, China. In addition, Tang et al. (2009) found that ozone volume fraction
increases at a rate of 1.1 ±0.5 ppbv $yr^{-1}$ during 2001–2006 in Beijing.

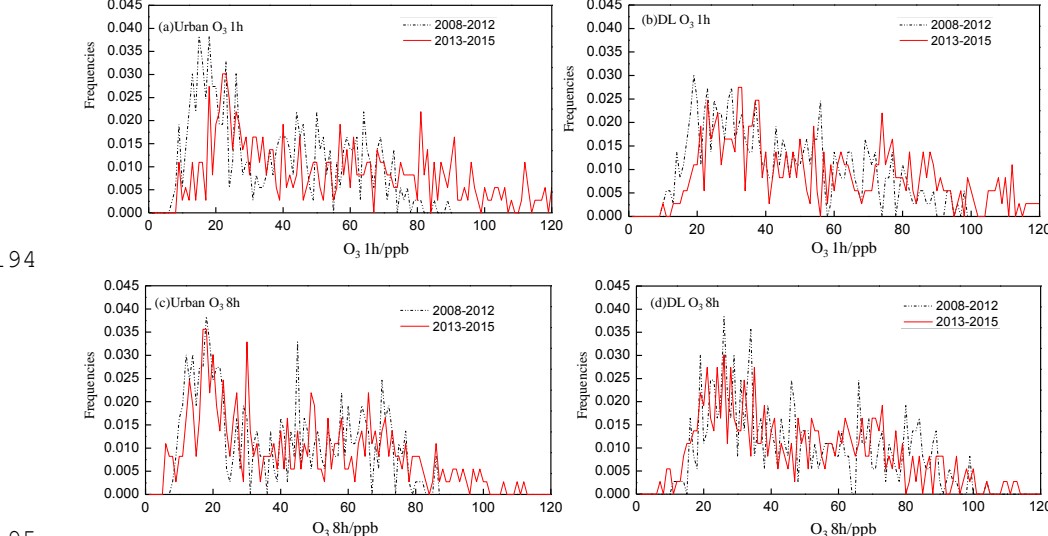



**Fig. 4** Frequency distribution of the ozone concentrations($O_3$1h and $O_3$8h) at DL and urbanBeijing for the periods of 2008–
2012 and 2013–2015

During 2008–2015, ratios between $NO_2$ and NO (lack of nitrogen oxide observed data
during 2004–2007, **Table 2**) increased significantly with AAGR of 0.46 (R=0.85, highly
relevant,95% confidence interval, two-tailed) and 1.87 (R=0.93,highly relevant,95%
confidence interval, two-tailed) at urban sites and DL background station respectively,
indicating ratios between $NO_2$ and NO were increasing larger at background station than
urban sites in Beijing. Average hourly concentrations of $NO_x$ were in the range of 29.21–
39.76 ppbv,11.85–13.91 ppbv at urban sites and DL background station, respectively. $NO_x$
concentration was in a downward trend with AAGR of −0.43 ppbv $yr^{-1}$ (R=0.32, real
relevant,95% confidence interval, two-tailed) and −0.21 ppbv $yr^{-1}$ (R=0.24, micro
relevant,95% confidence interval, two-tailed) at urban and DL background sites, respectively.
This study could not  further analyze the formation mechanism of ozone due to lack of
observed data for VOCs during 2008–2015, but research indicated the VOCs concentration
also declines in recent year in Beijing (Lu et al., 2010; Wang et al., 2015a). Wang et al.(2015a)
found mixing ratios of NMHCs measured at PKU university site decreased by 37% during
August increased by 28% from 2004 to 2012 and the measured NMHC/$NO_x$ ratios declined
by 14% during August from 2005 to 2012.Since,Non-methane hydrocarbons (NMHCs)





accounts for the vast majority of VOCs and plays a critical role in the photochemical
production of ozone, variations of VOCs should be further investigated for a thorough
understanding of ozone trends in the futher.
Ozone precursors (VOCs and $NO_x$) were decreasing, whereas average concentrations of
$O_3$8h were still increasing in urban Beijing, which may be caused by the particular sensitivity
regimes and other related factors. Although previous studies indicated the distribution of the
sensitivity regimes of $O_3$8h concentration was similar to that of $O_3$1h concentration (Zhang et
al., 2008), this distribution was a little different in urban Beijing (Shao et al., 2006). The
distribution of ozone sensitivity regimes is closely related to meteorological condition and
emission distribution (Zhang et al., 2008; Sillman, 1999). Ozone reactions are mainly VOCs-
sensitive and $NO_x$-sensitive in urban Beijing and suburban areas or more remote areas of
Beijing, respectively (Tang et al., 2009). In urban Beijing, a reduction of anthropogenic $NO_x$
could increase local ozone efficiently while a reduction of anthropogenic $NO_x$ in urban and
suburban areas could reduce ozone efficiently in downwind suburban areas.The other factors
caused by the increasing ozone concentration may be related to a significant increase in
regional tropospheric $NO_x$ concentrations, particularly in BTH area (Richter et al., 2005; Van
der A et al., 2006), or high concentrations of the regional zone and its precursors transport
(Parrish et al., 2014). Also,the rapid growth of population and industrialization have driven
substantial increases in ozone background concentrations in BTH area (Willem et al., 2015).
**Table 1** Statistics of $NO_2/NO$ in urban areas and at DL site during 2008–2015 in Beijing

| Parameter | | 2008 | 2009 | 2010 | 2011 | 2012 | 2013 | 2014 | 2015 |
|---|---|---|---|---|---|---|---|---|---|
| $NO_x$ /ppbv | Urban sites | 32.93±23.45 | 36.52±23.52 | 39.76±38.32 | 37.42±23.17 | 35.01±21.42 | 37.71±23.85 | 36.26±21.89 | 29.21±18.83 |
| | DL site | 13.91±11.95 | 16.30±14.17 | 15.12±9.33 | 17.95±14.59 | 16.35±14.45 | 17.66±16.72 | 14.42±11.95 | 11.85±10.74 |
| $NO_2/NO$ | Urban sites | 3.61±2.46 | 2.87±1.87 | 3.01±2.13 | 3.33±2.62 | 3.74±3.32 | 4.05±3.83 | 5.07±4.96 | 5.13±4.30 |
| | DL site | 4.89±3.01 | 3.68±2.24 | 4.27±2.15 | 4.75±2.17 | 8.12±6.67 | 8.41±5.48 | 10.83±6.88 | 12.18±10.97 |


### 3.2 Diurnal variations and regional transport

**Fig. 5** was the diurnal variation of ozone in urban Beijing and at DL station from May to
September during 2004–2015 and **Table 2** presented the statistics of ozone peaks at DL
station and urban sites from May to September during 2004–2015. In general,ozone
concentration at DL station was higher than that of urban sites, and peaks of ozone
concentration at DL station from May to September in different years was 1.01−1.56 times




that in urban sites.For the spatial distribution of ozone,it was lower in central urban area and
relatively higher in the northern and western area with good vegetation. Ozone has a lifespan
of several days; consequently, high ozone concentrations can be found in regions distant from
precursor emission sources (Seinfeld 2004; Kalabokas *et al.*, 2000), and several chemical
ozone destruction reactions existing in urban center, such as R2–R3, are absent in background
areas (Saitanis 2003; Pablo *et al.*, 2013). In Beijing, $NO_x$ concentration in the urban center of
the city was typically higher because of the large amount of vehicle population, which
consumed and titrated a certain amount of ozone. The ozone peaks at DL station from May to
September in different years was obviously 1-h behind than that of urban sites, which was
closely related to the regional ozone transport (R4). Most of the ozone was generated during
the transport of its precursors from emission districts to surroundings or background sites. In
summer, high temperature, strong solar radiation, low humidity, and small southwest wind in
Beijing strengthen photochemical pollution; moreover, ozone and its precursors, such as $NO_x$,
CO, and VOCs, are transported to the downwind area, hence the reduced ozone peak
concentration in downwind area (Carnero et al., 2010; Shan et al., 2010).
In addition,the difference of ozone peaks between DL station and urban sites was
significantly decreased from 18.20 ppbv to 2.72 ppbv during 2004–2010 and 2011–2015. This
change may be related to the expansion of urbanization of Beijing. With city expansion and
economic development, the district near DL stationwas urbanized and easily influenced by
anthropogenic emissions.Santini et al.(2010) found Beijing urban extent estimated from
Landsat data was from 1105km$^2$ to 4139km$^2$ between 2000 and 2009.Jacobson et al. (2015)
pointed that urbanization decreases the concentrations of many surface chemicals due to their
vertical dilution but increases near-surface ozone.

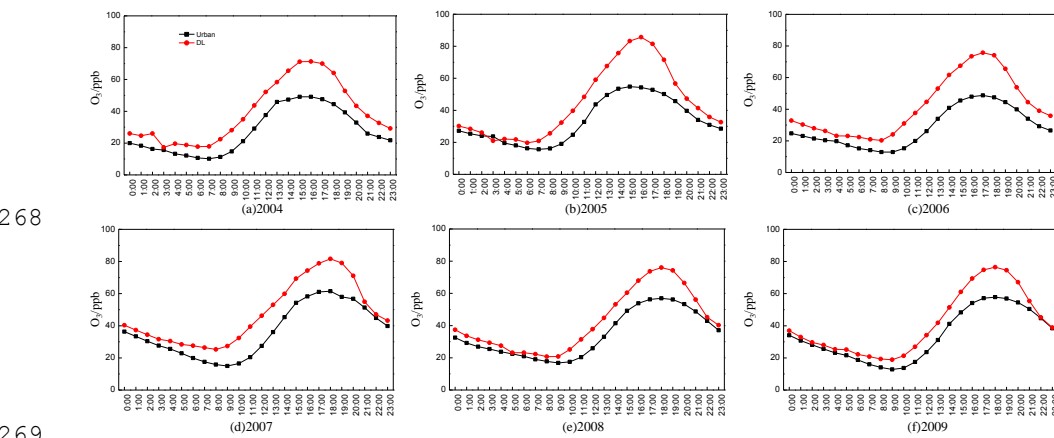







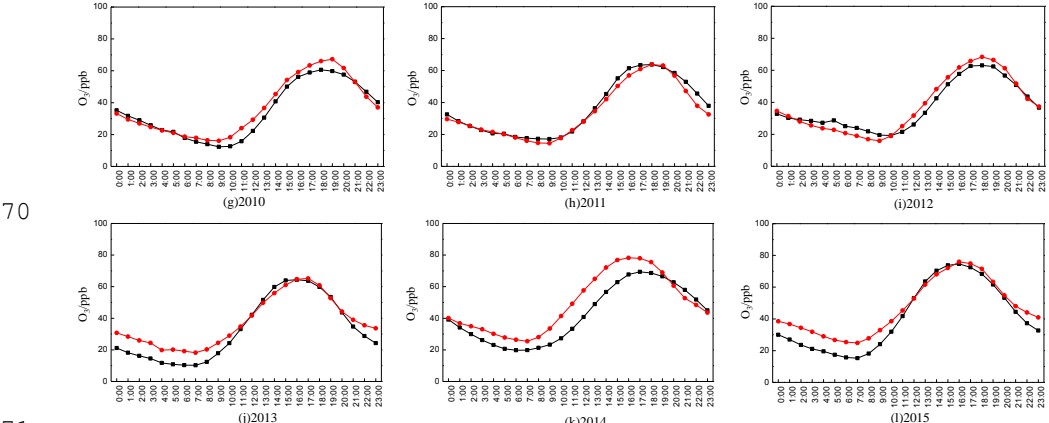

**Fig. 5** Diurnal variations of ozone in urban areas and at DL station from May to September during 2004–2015 in Beijing

**Table 2** Peak ozone concentration from May to September during 2004−2015 at DL station compared with the urban city of Beijing

| Parameter | 2004 | 2005 | 2006 | 2007 | 2008 | 2009 | 2010 | 2011 | 2012 | 2013 | 2014 | 2015 | 2004~2015 |
|---|---|---|---|---|---|---|---|---|---|---|---|---|---|
| delay hour/h | 1 | 1 | 0 | 0 | 0 | 0 | 1 | 1 | 0 | 1 | 0 | 1 | 0 |
| concentration difference /ppbv | 22.18 | 30.95 | 27.00 | 20.11 | 19.09 | 18.67 | 6.64 | 0.33 | 5.28 | 0.78 | 8.83 | 0.24 | 12.38 |

$$O_3 + OH \rightarrow HO_2 + O_2 \quad (R2)$$

$$O_3 + NO \rightarrow NO_2 \quad (R3)$$

$$NO_x + VOCs \rightarrow O_3 \quad (R4)$$

### 3.3 Emission reductions on ozone concentrations

Prevention and control of air pollution in BTH region and its surrounding areas, such as Shandong,Shanxi,and Inner Mongolia,effectively reduced the air pollutants' emission intensity caused by "human activities" and led to the Beijing blue during the parade on the 70th Victory Memorial Day for the Chinese People's War of Resistance against Japanese Aggression(Sep 3 military parade).Local enhanced reduction measures in Beijing were implemented firstly in S2(Aug 20~31,2015), and regional enhanced reduction measures in Beijing and its surrounding areas were implemented subsequently in S3(Sep 01~03,2015). Calculated average concentrations of CO, NO$_2$, and O$_3$ in S2 and S3 decreased by 31.48%, 43.97%, and 13.21% at urban sites, and by 20.93%, 57.10%, and 23.62% at DL site, respectively (**Table 3; Fig. 6**) compared with those in S1(Aug 01~19,2015) and S4(Sep 04~30,2015). After the implementation of enhanced reduction measures in the surrounding





area in S3, average concentrations of $NO_2$ and $O_3$ at the urban sites decreased by 3.95% and
8.05.0%, respectively compared with those in S2, and average concentrations of CO in S3
were close to those in S2. Overall, the enhanced reduction measures decreased most air
pollutants in Beijing significantly and effectively.

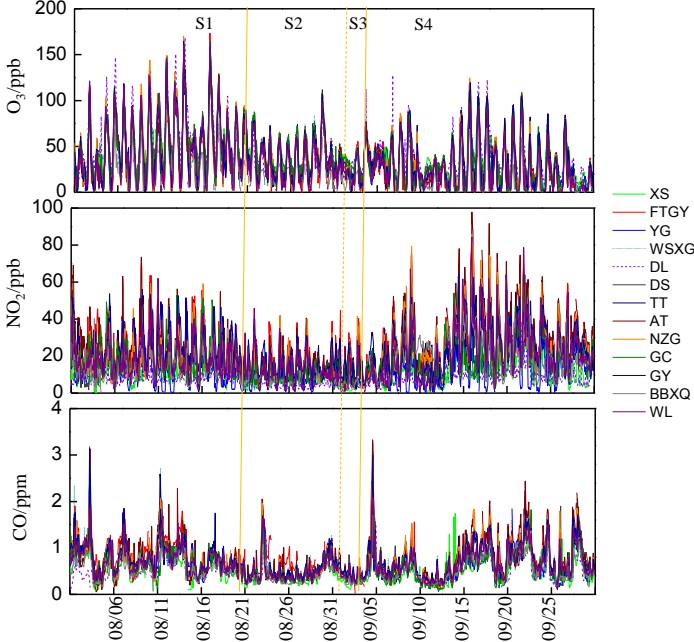

**Fig. 6** Concentrations of main air pollutants in urban Beijing from August 01 to September 30 in 2015.
**Table 3** Average concentrations of main air pollutants at urban sites at four stages in Beijing in 2015

| Pollutants | Stage | XS | FTGY | YG | WSXG | DS | TT | AT | NZG | GC | GY | BBXQ | WL | DL |
|---|---|---|---|---|---|---|---|---|---|---|---|---|---|---|
| | S1 | 52.96 | 50.47 | 48.86 | 51.41 | 46.89 | 48.37 | 50.17 | 53.21 | 52.23 | 50.88 | 34.26 | 51.33 | 56.37 |
| | S2 | 38.02 | 34.67 | 38.89 | 39.73 | 35.14 | 36.14 | 37.78 | 39.22 | 42.64 | 36.92 | 17.66 | 32.97 | 33.55 |
| $O_3$/ppb | S3 | 36.59 | 34.61 | 33.72 | 34.38 | 31.36 | 33.23 | 35.54 | 34.43 | 37.46 | 33.97 | 13.40 | 30.47 | 36.93 |
| | S2~S3 | 37.31 | 34.64 | 36.31 | 37.05 | 33.25 | 34.68 | 36.66 | 36.83 | 40.05 | 35.44 | 15.53 | 31.72 | 35.24 |
| | S4 | 33.66 | 31.82 | 28.95 | 28.09 | 27.49 | 25.23 | 32.30 | 29.77 | 28.69 | 31.97 | 16.78 | 26.69 | 35.90 |
| | S1 | 9.89 | 25.72 | 12.17 | 19.60 | 17.95 | 19.72 | 24.95 | 19.42 | 19.04 | 24.64 | 15.00 | 18.32 | 8.63 |
| | S2 | 6.02 | 16.38 | 6.59 | 13.20 | 12.89 | 14.87 | 14.98 | 13.16 | 10.53 | 15.17 | 7.55 | 12.90 | 3.71 |
| $NO_2$/ppb | S3 | 5.78 | 16.45 | 7.67 | 12.98 | 12.69 | 13.85 | 13.60 | 12.33 | 11.50 | 14.42 | 7.16 | 9.70 | 3.97 |
| | S2~S3 | 5.90 | 16.42 | 7.13 | 13.09 | 12.79 | 14.36 | 14.29 | 12.74 | 11.01 | 14.79 | 7.35 | 11.30 | 3.84 |
| | S4 | 11.68 | 27.32 | 14.35 | 22.42 | 25.06 | 25.01 | 31.52 | 26.26 | 20.27 | 23.90 | 21.34 | 24.27 | 9.26 |
| | S1 | 0.58 | 0.98 | 0.73 | 0.85 | 0.78 | 0.80 | 0.79 | 0.78 | 0.74 | 0.82 | 0.57 | 0.75 | 0.54 |
| | S2 | 0.35 | 0.73 | 0.47 | 0.50 | 0.45 | 0.46 | 0.49 | 0.45 | 0.45 | 0.49 | 0.34 | 0.42 | 0.38 |
| CO/ppm | S3 | 0.36 | 0.63 | 0.52 | 0.58 | 0.55 | 0.56 | 0.57 | 0.51 | 0.49 | 0.56 | 0.38 | 0.50 | 0.44 |
| | S2~S3 | 0.36 | 0.68 | 0.50 | 0.54 | 0.50 | 0.51 | 0.53 | 0.48 | 0.47 | 0.52 | 0.36 | 0.46 | 0.41 |
| | S4 | 0.50 | 0.74 | 0.61 | 0.78 | 0.76 | 0.74 | 0.82 | 0.74 | 0.58 | 0.74 | 0.57 | 0.68 | 0.50 |




In order to eliminate the influence of meteorological factors, we counted the observed
variations of meteorological elements from Aug 20 to Sep 05 at GXT station in Beijing
between 2010 and 2015 (We only collected meteorological data for the past five years).From
**Table 4,**the temperature and wind speed at ground which can affect ozone concentration
directly changed slightly during the study periods. Average temperature was fluctuating
between 23.6~24.3℃ and it was  was lowest in 2011 and highest in 2013. Average of wind
speeds was fluctuating between 1.4~1.9 m s$^{-1}$,suggesting the atmosphere was generally stable
in August in Beijing.Average relative humidity and suface pressure was also possessing the
same characteristics.Frequency of the north wind at 850hPa directly affects vertical diffusion
of ozone.Average frequency of the north wind at 850hPa was between 12.1% and 41.2% and
it changed to 24.3% from Aug 20 to Sep 05,2015 indicating the atmosphere was also
relatively stable in the vertical direction It is reported that both sunshine hours and visibility
in BTH region have been decreasing in the past decades (Yang et al., 2009; Zhao et
al.,2011).But we could not detect the significant decrease of sun-shine hours due to lack of
observed meteorological data. Briefly, we suppose that the meteorological elments might play
only a minor role in the ozone concentration changes in Beijing, and then focus our discussion
on the effects of regional emission reduction measures.

**Table4.**Average meteorological elements from Aug 20 to Sep 05 at GXT station in Beijing between 2010 and 2015.

| Year | RH/% | Suface Pressure/hPa | T/℃ | Speed/(m s$^{-1}$) | Frequency of the north wind at 850hPa /% | T at 850hPa /℃ |
|---|---|---|---|---|---|---|
| 2010 | 70.7 | 1008.0 | 24.1 | 1.6 | 20.6 | 16.0 |
| 2011 | 73.2 | 1007.4 | 23.6 | 1.7 | 12.1 | 16.1 |
| 2012 | 68.6 | 1006.8 | 23.8 | 1.9 | 41.2 | 15.9 |
| 2013 | 66.2 | 1005.8 | 24.3 | 1.9 | 25 | 16.1 |
| 2014 | 67.6 | 1006.5 | 24.2 | 1.7 | 14.7 | 16.7 |
| 2015 | 69.1 | 1009.9 | 24.1 | 1.4 | 24.3 | 13.9 |


We furher analyzed the year-on-year comparisons of diurnal variations of ozone from
2004 to 2015 during the same periods of air quality assurance stages(S2 and S3)(**Fig.7ab**).
Comparing the ozone peaks in urban Beijing during S2 and S3 stages between 2004-2014 and
2015(the year of taking emission reduction measures to ensure the regional air quality), the
ozone peak in 2015 was 3h earlier and  0.91 ppbv lower compared to the average ozone peaks
during the period of 2004-2014.Whereas comparing the diurnal variations of ozone at DL site
during S2 and S3 stages between 2004-2014 and 2015, the ozone peak in 2015 was 10.98
ppbv lower and 2 h earlier compared to that during the period of 2004-2014.The ozone peaks
between urban sites and DL site were much closer (only about 0.48ppbv) after the    a





reduction of anthropogenic emissions in 2015.The earlier ozone peaks indicated the
approximate photochemical equilibrium(R5-R7) of O$_3$,NO and NO$_2$ was moved up in Beijing
during the day due to regional emission reductions.Therefore,a reduction of anthropogenic
emissions such as VOCs and NO$_x$ in urban areas made the ozone peaks decrease
significantly and appear 2~3h earlier compared to the scenarios of no emission reductions
which was a very interesting  phenomenon and first found in Beijing and it could also reduce
ozone concentration efficiently especially at background sites or  downwind areas by the
weakened  regional  transport  which  was  coincident  with  the  study  of  Seinfeld
(Seinfeld ,2006) .

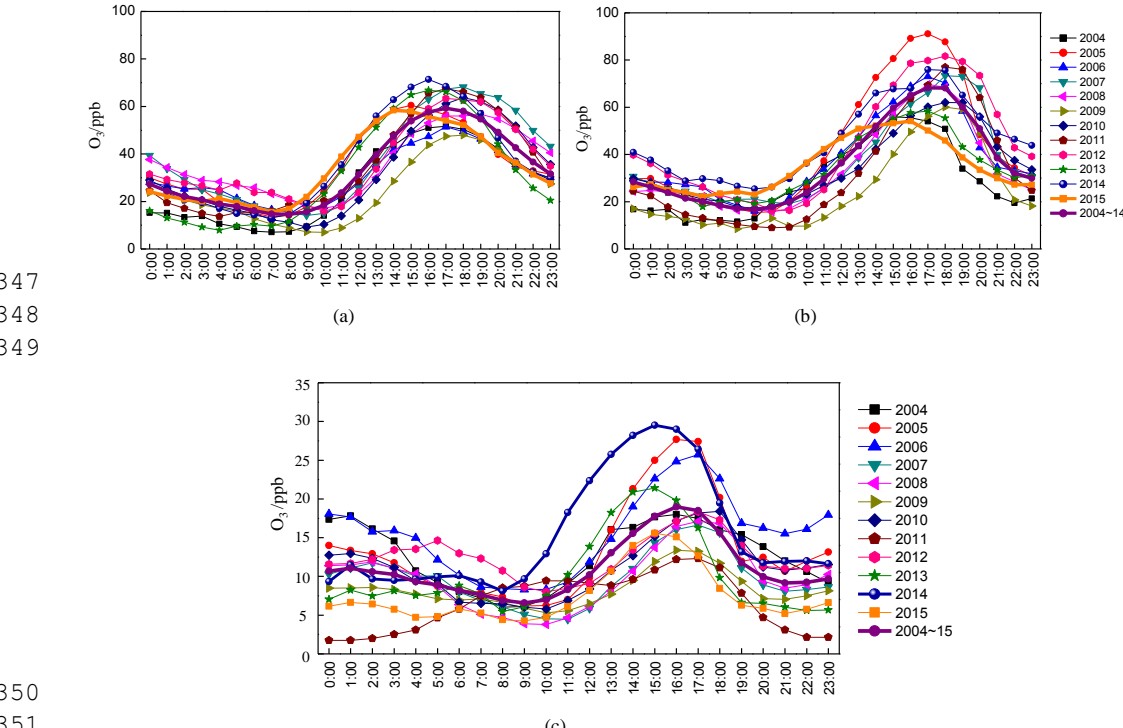

**Fig. 7** Diurnal variations of ozone in urban  Beijing (a) and at DL station (b) during S2 and S3 stages from 2004  to 2015
compared to that of APEC meeting air quality assurance period(Nov 1~Nov 11,2014) in  urban  Beijing (c) from 2004  to

354 2015


NO$_2$+ $hv$ $\rightarrow$ NO+ O          (R5)
O$_2$+O$\rightarrow$O$_3$              (R6)
O$_3$+NO$\rightarrow$NO$_2$+O$_2$          (R7)



Compared to the variations of ozone during  Asia-Pacific Economic Cooperation(APEC)
meeting air quality assurance period(Nov 1~Nov 11,2014;**Fig7c**) in  urban  Beijing,the ozone
peak in 2014 was 1h earlier and  about 9.6 ppbv higher compared to the average ozone peaks
during  the  period  of 2004-2015 although Beijing and its surrounding areas also adopted
different levels of emission reduction measures.
As  we  all  know,$NO_x$  and  VOCs  emission  control  can  considerably  affect  the  $O_3$
concentration, but ozone generation is not a simple linear relationship with its precursors
(Sillman, 1999).Ozone pollution is mainly concentrated in summer, and biogenic emissions
accounted for a majority of the total VOCs. Therefore, the emission reduction of VOCs via
anthropogenic measures connot make it  higher than that of $NO_x$.To ensure the air quality
during  the  military  parade(S2 and S3 stages) in 2015, $NO_x$  and  VOCs  emission  control  in
Beijing and its surrounding areas lasted for almost a month, and VOCs emission control
measures  was  much  stricter  than  $NO_x$(MEP, 2015); thereby, ensuring the reduction of VOCs
emission(45%) is higher than that of $NO_x$(30%).While for the temporal distribution of ozone
during APEC meeting air quality assurance period, regional VOCs emission(about 30%) was
equal  to   that of $NO_x$( about 30%)(MEP, 2015) and it was easily affected by the  relatively
unfavorable diffusion conditions in Autumn in Beijing which lead to the concentrations of
$NO_x$ and CO two times larger than that of Sep 3 military parade period.So different emission
reduction  ratio  between  $NO_x$  and  VOCs   and  different  weather  conditions  led  to  different
VOC(ppbvC)/NOx(ppbv)ratios during Sep 3 military parade period and APEC meeting.If the
$NO_x$ levels are so high that it is not consumed before the end of the day,then ozone is VOCs
sensitive,  and  decreasing  $NO_x$  would  cause  increased  ozone   formation  during  APEC
meeting.This phenomenon of concentrations of most of the air pollutants decreased, whereas
concentrations of ozone  increased during APEC meeting period which was consistent  with
the study of Wang (Wang et al,2015b).Whereas,the higher VOCs emission  reduction caused
the slight decrease in urban area but significant decrease at downwind DL background station
during Sep 3 military parade period.
Above all,success of air quality protection during the Sep 3 military parade proved that
the  current  governance  policy  is  correct  and  far-sighted.  Moreover,  ozone  pollution  is
typically a regional rather than a local issue. Thus, in the future, clean air action plan in
Beijing  should  be  implemented  on  the  basis  of  the  lessons  from  regional  air  pollution
prevention  and  control  mechanism  to  promote  the  continuous  improvement  of  regional  air
quality unswervingly and jointly. Combined with multiple observation stations of ozone in a



long period, the numerical models should also be combined to further analyze the ozone
formation, so as to develop effective ozone pollution control measures.

**4 Conclusions**
4.1 Annual concentration of daily maximum 1 h ozone ($O_3$1h) was all increasing at urban
sites(1.79 ppbv yr$^{-1}$) and DL background station(2.05 ppbv yr$^{-1}$) while daily maximum 8 h
average ozone concentration($O_3$8h) was increasing in urban area(1.14 ppbv yr$^{-1}$) and slightly
decreasing at DL background station(-0.47 ppbv yr$^{-1}$) from 2004 to 2015 due to different
ozone sensitivity regimes and ratios of $NO_2$/NO.
4.2 Diurnal variation of ozone peaks obtained at the downwind DL station were about 1 h
later than that of the urban area from May to October in different years and concentration of
ozone at downwind background station was much higher than that of urban sites. Moreover,
difference of ozone peaks between urban sites and DL background station was significantly
becoming smaller in recent years, which may be related to regional ozone transport and the
expansion urbanization of Beijing.
4.3 Based on the joint efforts of regional air pollution prevention and control,Beijing achieved
Sep 3 military blue. Average concentrations of CO, $NO_2$, and $O_3$ in S2(Aug 20~31,2015) and
S3(Sep 01~03,2015) decreased by 31.48%, 43.97%, and 13.21% in urban sites, and by
20.93%, 57.10%, and 23.62% at DL site, respectively compared with those in S1(Aug
01~19,2015) and S4(Sep 04~30,2015).A reduction of local anthropogenic emissions such as
VOCs and $NO_x$ could reduce ozone efficiently especially in downwind areas of Beijing and
made the ozone peaks decrease significantly and appear 2~3h earlier compared to the
scenarios of no emission reductions. Compared to the increasing ozone during APEC
period,to decrease the ozone concentration in Beijing, emissions of VOCs should be reduced
larger than that of $NO_x$ in Beijing and the policy of regional air pollution joint prevention and
control should still be promoted unswervingly and jointly in the futher.

**Acknowledgments**
This study was supported by the Commonwealth Project of the Ministry of
Environmental Protection (NO.201409005) and the National Key Technology R&D Program
(2014BAC23B03). For detailed data,please see website http://zx.bjmemc.com.cn/ or send an
email to15001195306@163.com.
The English in this document has been checked by at least two professional editors, both
native speakers of English.



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
