# Peer review of "Characteristics of Ground Ozone Concentration over Beijing from 2004 to 2015: Trends, Transport, and Effects of Reductions"

_Atmospheric Chemistry and Physics, 2016_

## Referee Comment (RC1) · Anonymous Referee #1 · 27 Oct 2016

This manuscript presents discussions of surface ozone (O3) characteristics based on the hourly monitoring data during 2004-2015 in Beijing, including the difference of O3 production between local and rural area in Beijing, evaluation of the O3 before, during, and after Sep 3 military parade in 2015, and the O3 impact of emissions reductions during the APEC period. The long-term observed data used in the manuscript provides a detailed description of the O3 variation in Beijing in recent 10 years. However, the manuscript lacks novelty, scientific analyses and discussions. Several similar papers focusing on the O3 observation in Beijing have been published (e.g., Tang et al., 2009; Wang et al., 2012; Zhang et al., 2014; Ma et al., 2016), so what is new in the manuscript? The reviewer would like to recommend a major revision before publication.

[Figure]

Major comments:

1. P6, Line 135-147, O31h increased with an annual concentration growth rate (AAGR) of 1.79 ppbv.yr-1 and a higher growth rate of 2.84 ppbv.yr-1 during May to September from 2004 to 2015 at the urban area, but at DL background station, O31h showed an increasing trend with an AAGR of 2.05 ppbv.yr-1 and the growth rate during May-September was 0.14 ppbv.yr-1, which is much smaller than the AAGR. More detailed analysis is needed to make comparison of O3 concentrations between the urban and rural area. The authors just present a description of the distribution of O31h and O38h concentrations during different periods at twelve sites in the urban Beijing and DL background site. In addition, the authors should provide the reason for choosing the O3 concentration from May to September.

Furthermore, Line 199-233, the authors provide the analysis of O3 precursors to illustrate the high concentration in Beijing local area, but fail to present comprehensive explanations to compare the characteristics of O3 production between local Beijing and DL site.

2. P11, Line 260-261, "In addition, the difference of ozone peaks between DL station and urban sites was
significantly decreased from 18.20 ppbv to 2.72 ppbv during 2004–2010 and 2011–2015. "" Why is the years divided into 2004-2010 and 2011-2015? The time division in the manuscript is confusing, for example, Line 161-164, the authors divided the year of 2004-2015 into three parts: 2004-2007, 2008-2012, and 2013-2015. More detailed explanations for the time division are needed in the paper.

3. The section 3.3 only confirms the emission reduction implemented by the government. Although the authors have compared the effect of emissions reductions between Sep 3 military parade period and APEC meeting, sufficient scientific analyses of the difference between them are not presented, such as the characteristics of O3 production, meteorology etc. The result "So different emission reduction ratio between NOx and VOCs and different weather conditions led to different VOC/NOx ratios during Sep

3 military parade period and APEC meeting." in Line 377 is too general.

4. In the abstract and conclusion, the authors emphasize that the difference of annual O3 concentration at urban and rural area was attributed to different sensitivity regimes, but the manuscript does not provide sufficient analyses to prove the statement.

Minor comments

5. Line 185, revise " increased " to " increase ".

6. Line 200, " Table 2 " should be " Table 1 ".

7. Line 263, " stationwas " should be revised as " station was ".

8. Line 299, " 8.05.0% " should be changed to " 8.05% ".

9. Line 337, " a " should be deleted.

10. Pay more attention to the format of the reference.

———————————————

---

## Referee Comment (RC2) · Anonymous Referee #3 · 22 Nov 2016

This paper presents analysis and discussions of a decade-long trend of the surface ozone concentration in Beijing during 2004-2015 based on hourly monitoring data. It also discusses the effects of transport and emission reduction on the surface ozone concentrations. Although it presents an interesting and valuable data set, the manuscript is poorly organized, poorly presented and poorly written. The analysis and discussion are very unclear and confusing, and no much valuable scientific information on the O3 pollution in Beijing is derived. A dramatic revision is needed before it is considered for publishing.

---

## Author Comment (AC1) · 1 Jan 2017

The logical structure of this study is to explain the change trend of ozone firstly, followed by the analysis of the effects of the reduction measures on the ozone concentration which is supposed to one of the classical logic structures. The modified text has been marked in red font in the revised article.

1. In Beijing, the nonattainment days of ozone is mainly concentrated in the months of May to September and the comparisons of trend analysis between nonattainment days and reaching standard days were very meaningful in some mega cities especially Beijing. (http://www.bjepb.gov.cn/bjepb/323474/324034/324735/index.html). Restricted by P(Ox=O3+NO2) calculation and observation,we fail to present comprehensive explanations to compare the characteristics of O3 production between local Beijing and DL site. But according to the analysis (Zhang et al.,2014), VOCs and NOx both decreased between 2006 and 2011 and the decrease in VOCs reactivity (−5%yr−1) was slightly larger than the decrease in NOx (−4%yr−1), leading to a slight decrease in P(Ox). The sunshine hours and visibility are also the important factors influencing O3 production. Hence, variations of P(Ox) need to be further investigated for a better understanding of ozone trends. 2. The Mann–Kendall trend detection test(Kendall, 1975) has been commonly applied to assess the significance of monotonic trends in ozone pollution data time series and we found the year 2013 was an intercept break point. In 2008, Beijing held the Olympic Games and the ozone in Beijing decreased significantly.Thegood air quality promote the awareness of environmental protection to take various common emission reduction mearures in the next few years. In 2013,the state implemented a new environmental air quality standards, marking the work of environmental protection into a new stage. We divided the available the year of 2004-2015 of zone data into three parts: 2004-2007, 2008-2012, and 2013-2015.

3. The previous studies had proved that HCHO and NO2 from the OMI serve as appropriate indicators for in situ observations of total reactive nitrogen and VOCs (Liu et al.,2016;Duncan et al.,2010). OMI tropospheric HCHO/NO2 Ratio (Ratio)<1 represents PO3 reduces with diminishing in VOCs (VOC-limited conditions), and Ratio>2 represents NOx-limited conditions. When the ratio is between 1 and 2 indicates a transition regime (mixed VOCs-NOx-limited regime) where the instantaneous PO3 could be affected by both VOCs and NOx emissions. Three episodes are separately defined in this study: 1st episodes is defined as the period of Parade (from August 20th to September 3rd 2015); 2nd and 3rd episodes are defined as the "pre-Parade" from August 1st to 19th and the "post-Parade" from September 4th to 30th. Similarly, the three episodes of APEC are respectively defined as pre-APEC period (from October 15th to 30th), APEC period (from November 1st to 12th) and post-APEC period (from November 13th to 30th). With a series of strict emission control measures during Parade, the ratio retrieved from OMI had changed from 1.70 to 3.72. It means the PO3 conditions had also changed from mixed VOCs-NOx-limited to a predominantly NOx-limited condition due to the sharp drop of NO2 during Parade periods. After the strict emission control measures, the NO2 returned to relatively high values as pre-Parade and the ratio was also diminished (Ratio = 0.90, < 1), which indicated the PO3 was turned into a VOCs-limited condition during post-Parade. To ensure the air quality during the military parade in 2015, NOx and VOCs emission control in Beijing and its surrounding areas lasted for almost a month, and VOCs emission control measures was much stricter than NOx; thereby, ensuring the reduction of VOCs emission (45%) is higher than that of NOx (30%)(MEP, 2015). Higher ratios by emission control measures during Parade were not only work effectively for NO2 pollution control patterns but also effective for O3 controlling. 4. Compared with pre-APEC, the ratio was changed from around 1.21 (VOCs-limited and mixed VOCs-NOx-limited) to around 1.60 (mixed VOCs-NOx-limited and NOx-limited) in Beijing during APEC. NO2 and HCHO had a certain reduction during APEC which should lead the O3 diminishing. Conversely, the ozone concentration was increasing compared to pre-APEC. Regional VOCs emission (about 30%) was equal to that of NOx (about 30%) (MEP, 2015) during APEC periods and it was easily affected by the relatively unfavorable diffusion conditions in Autumn in Beijing which lead to the concentrations of NOx and CO two times larger than those of the 2015 Grand Military Parade. So different emission reduction ratios between NOx and VOCs and different weather conditions led to different VOCs (ppbv)/NOx (ppbv)ratios during the 2015 Grand Military Parade periods and APEC meeting. These results indicated that emission controls in this case maybe not strict enough or worked well to lessen the levels of ozone. This phenomenon of concentrations of most of the air pollutants decreased, whereas concentrations of ozone increased during APEC meeting period which was consistent with the study of Wang (Wang et al,2015b; Liu et al.,2016).

Minor comments (The authors have revised the following errors.) 5. Line 185, revise " increased " to " increase ". 6. Line 200, " Table 2 " should be " Table 1 ". 7. Line 263, " stationwas " should be revised as " station was ". 8. Line 299, " 8.05.0% " should be changed to " 8.05% ". 9. Line 337, " a " should be deleted. 10. Pay more attention to the format of the references. This manuscript has been further edited for language by Essaystar (http://essaystar.com/Service.html), acompany dedicated to helping international researchers publish their findings in the best English language journals possible.

Please also note the supplement to this comment:
http://www.atmos-chem-phys-discuss.net/acp-2016-508/acp-2016-508-AC1-supplement.pdf

**Supplement:**

[revised manuscript text omitted]

Air quality security programs were implemented from August 20 to September 3 in 2015

to guarantee the air quality for the parade on the 70th Victory Memorial Day for the Chinese

People's War of Resistance against Japanese Aggression (the 2015 Grand Military Parade).

Chinese government established numerous emission reduction measures, such as reducing coals, industrial adjustment, joint prevention measures, and limitation of vehicles (particularly heavy-duty buses and trucks from outside Beijing, and odd-even license plate policy on roads within urban Beijing). As regional emission reduction measures can not be copied and costs a lot of manpower and material resources, it offers a precious opportunity to study the changes in ozone and its precursors during the period of air quality assurance.

This paper aims to investigate the temporal trends of $O_3$1h and $O_3$8h in different sites in

Beijing and verify the importance of ozone transport. Also, we evaluated the changes on ozone concentration after the reduction measures during the 2015 Grand Military Parade in

2015.

**2 Materials and methods**

**2.1 Site distribution**

Beijing is located at 115.7 °–117.4 °E, 39.4 °–41.6 °N. This area is at the northwest edge of the North China Plain and close to the edge of the semi-desert zone. Its terrain exhibits a dustpan shape, and it is surrounded by mountains in three directions. The average altitude of

Beijing is 43.5 m, and the general altitude of mountains is in the range of 1 000–1 500 m, which is not conducive to air pollutants diffusion. The total area of Beijing is 16410.54 $km^2$, in which 62% are mountains. Its total forest coverage in the plain region is about 15%, which is lower than that in whole city (38%). Beijing exhibits a temperate continental monsoon climate, where it is hot and rainy in summer and cold and dry in winter. Over the past decade, the annual averaged rainfall is less than 450 mm, 80% of which is concentrated in June, July, and August (BJEPB,2014; Beijing Statistics Bureau, 2014).

As the capital of China, the air quality monitoring network in Beijing is more advanced than other regions of China (BJEPB, 2014). In 2001, an air quality monitoring network that obtains 35 stations was established by the Beijing Municipal Environmental Monitoring

Center (BJMEMC, http://zx.bjmemc.com.cn/, **Fig. 1**). The 35 stations cover all districts that contain different environment types defined by regional background, such as suburbs, city, and residential. Twelve monitoring sites (DL, DS, GY, TT, WSXG, AT, NZG, WL, GC, SY,

CP, HR) in urban area and one background station DL were selected in Beijing and used in this study. DL station (116.22 °E, 40.29 °N, about 45 km northwest of Tiananmen square) is the background station of World Meteorological Organization Environmental Monitoring center in China and has conducted air pollutant monitoring work for decades.Meteorological sounding data in Beijing at Guanxiangtai station (GXT,54511, 116.46 °E, 39.80 °N) were downloaded from the Department of Atmospheric Science, College of Engineering, University of Wyoming (http://weather.uwyo.edu/upperair/sounding.html).

In this study,we retrieved $NO_2$ and HCHO VCDs from the OMI products in urban Beijing (GY site, 116.33 °E, 39.93 °N) and combined the corresponding ratios to analyze the chemical sensitivity of $PO_3$ (ozone production rate) during both Parade and APEC periods.

[Figure]

**Fig. 1** Distribution and classification of observation sites in Beijing

**2.2 Monitoring instruments**

Ozone are mornitored by the 49C ozone analyzer instruments produced by Thermo Fisher Corporation (USA). The minimum limit of ozone analyzer instrument is $1 \times 10^{-9}$, and the zero and cross drifts are 0.4%/24 h and ±1% /24 h, respectively. An ozone calibrator (49IPS) traceable to the Standard Reference Photometer maintained by the WMO World Calibration Center was used to calibrate the ozone analyzers. Ozone monitoring instrument at each station had a zero cross calibration every three days, precision audit every three month, and an accuracy check every six months to ensure the accuracy of ozone monitoring in Beijing. Thermo Fisher 42C NO–$NO_2$–$NO_x$ analyzer was used to monitor NO and $NO_2$

concentrations with a limit of $0.05 \times 10^{-9}$, zero drift of $0.025 \times 10^{-9}/24$ h, and span drift of $\pm 1\%/24$h. Operation procedure strictly followed the "The Specification of Environmental Air Quality Automatic Monitoring Technology" (HJ/T193-2005, http://kjs.mep.gov.cn/hjbhbz/bzwb/dqhjbh/jcgfffbz/200601/t20060101_71675.htm), and the equipments were regularly calibrated and maintained by technicians.

**3 Results and Discussion**

**3.1 Variations trends**

[revised manuscript text omitted]

Beijing, a reduction of anthropogenic $NO_x$ could increase local ozone efficiently while a reduction of anthropogenic $NO_x$ in urban and suburban areas could reduce ozone efficiently in downwind suburban areas. Except for different regimes,the other factors caused by the increasing ozone concentration may be related to a significant increase in regional tropospheric $NO_x$ concentrations, particularly in BTH area (Richter et al., 2005; Van der A et al., 2006), or high concentrations of the regional zone and its precursors transport (Parrish et al., 2014). Also,the rapid growth of population and industrialization have driven substantial increases in ozone background concentrations in BTH area (Willem et al., 2015). Restricted by P(Ox=$O_3$+$NO_2$) calculation and observation,we fail to present comprehensive explanations to compare the characteristics of $O_3$ production between local Beijing and DL site. But according to the analysis (Zhang et al.,2014), VOCs and NOx both decreased between 2006 and 2011 and the decrease in VOCs reactivity ($-5\%yr^{-1}$) was slightly larger than the decrease in NOx ($-4\%yr^{-1}$), leading to  a slight decrease in P(Ox). The sunshine hours and visibility are also the important factors influencing $O_3$ production. Hence, variations of P(Ox) need to be further investigated for a better understanding of ozone trends.

[revised manuscript text omitted]

The previous studies had proved that HCHO and $NO_2$ from the OMI serve as appropriate
indicators for in *situ* observations of total reactive nitrogen and VOCs (Liu et al.,2016;Duncan
et al.,2010). OMI tropospheric HCHO/$NO_2$ Ratio (Ratio)<1 represents $PO_3$ reduces with
diminishing in VOCs (VOC-limited conditions), and Ratio>2 represents NOx-limited
conditions. When the ratio is between 1 and 2 indicates a transition regime (mixed VOCs-
NOx-limited regime) where the instantaneous $PO_3$ could be affected by both VOCs and NOx
emissions. Three episodes are separately defined in this study: 1st episodes is defined as the
period of Parade (from August 20th to September 3rd 2015); 2nd and 3rd episodes are defined
as the "pre-Parade" from August 1st to 19th and the "post-Parade" from September 4th to
30th. Similarly, the three episodes of APEC are respectively defined as pre-APEC period
(from October 15th to 30th), APEC period (from November 1st to 12th) and post-APEC
period (from November 13th to 30th).

With a series of strict emission control measures during Parade, the ratio retrieved from
OMI had changed from 1.70 to 3.72. It means the $PO_3$ conditions had also changed from
mixed VOCs-NOx-limited to a predominantly NOx-limited condition due to the sharp drop of
$NO_2$ during Parade periods. After the strict emission control measures, the $NO_2$ returned to
relatively high values as pre-Parade and the ratio was also diminished (Ratio = 0.90, < 1),
which indicated the $PO_3$ was turned into a VOCs-limited condition during post-Parade. To
ensure the air quality during the military parade in 2015, $NO_x$ and VOCs emission control in
Beijing and its surrounding areas lasted for almost a month, and VOCs emission control
measures was much stricter than $NO_x$; thereby, ensuring the reduction of VOCs emission
(45%) is higher than that of $NO_x$ (30%)(MEP, 2015). Higher ratios by emission control
measures during Parade were not only work effectively for $NO_2$ pollution control patterns but
also effective for $O_3$ controlling.

Compared with pre-APEC, the ratio was changed from around 1.21 (VOCs-limited and
mixed VOCs-NOx-limited) to around 1.60 (mixed VOCs-NOx-limited and NOx-limited) in
Beijing during APEC. $NO_2$ and HCHO had a certain reduction during APEC which should lead the O₃ diminishing. Conversely, the ozone concentration was increasing compared to pre-APEC. Regional VOCs emission (about 30%) was equal to that of $NO_x$ (about 30%) (MEP, 2015) during APEC periods and it was easily affected by the relatively unfavorable diffusion conditions in Autumn in Beijing which lead to the concentrations of $NO_x$ and CO two times larger than those of the 2015 Grand Military Parade. So different emission reduction ratios between $NO_x$ and VOCs and different weather conditions led to different VOCs (ppbv)/$NO_x$ (ppbv)ratios during the 2015 Grand Military Parade periods and APEC meeting. These results indicated that emission controls in this case maybe not strict enough or worked well to lessen the levels of ozone. This phenomenon of concentrations of most of the air pollutants decreased, whereas concentrations of ozone increased during APEC meeting period which was consistent with the study of Wang (Wang et al,2015b; Liu et al.,2016).

[Figure]

**Fig. 8** The variations of averaged Ratio of three periods during Parade (A) and during APEC (B) at Beijing urban sites

Above all, success of air quality protection during the 2015 Grand Military Parade proved that the current governance policy is correct and far-sighted. Moreover, ozone pollution is typically a regional rather than a local issue. Thus, in the future, clean air action plan in Beijing should be implemented on the basis of the lessons from regional air pollution prevention and control mechanism to promote the continuous improvement of regional air quality unswervingly and jointly. Combined with multiple observation stations of ozone in a long period, the numerical models should also be combined to further analyze the ozone formation, so as to develop effective ozone pollution control measures.

**4 Conclusions**

Although Beijing local government has spent considerable efforts in Beijing to improve air quality and concentrations of the main air pollutants declined significantly. Resluts showed that nnual averaged concentration of daily maximum 1 h ozone (O₃1h) was all increasing at urban sites (1.79 ppbv $yr^{-1}$) and DL background station (2.05 ppbv $yr^{-1}$) while daily maximum 8 h averaged ozone concentration ($O_38h$) was increasing in urban area (1.14

ppbv $yr^{-1}$) and slightly decreasing at DL background station (-0.47 ppbv $yr^{-1}$) from 2004 to

2015 due to different ozone sensitivity regimes and ratios of $NO_2/NO$.

Diurnal variations of ozone peaks obtained at the downwind DL station were about 1

hour later than those of the urban sites from May to October in different years and concentrations of ozone at downwind background station were much higher than those at urban sites. Moreover, differences of ozone peaks between urban sites and DL background station were becoming significantly smaller in recent years, which may be related to regional ozone transport and the expansion urbanization of Beijing.

During several major activities held in Beijing such as the Asia-Pacific Economic

Cooperation (APEC) Summit in 2014 and the Parade on the 70th Victory Memorial Day for the Chinese People's War of Resistance against Japanese Aggression in 2015,Beijing and its neighboring cities implemented numerous control strategies, including the suspension of factory operations and odd-and-even license plate rules. A reduction of anthropogenic emissions such as VOCs and $NO_x$ could reduce ozone efficiently especially in downwind areas of Beijing and made the ozone peaks decrease significantly and appear 2~3hours earlier compared to the scenarios of no emission reductions.

On the basis of the discussion and analyses, several recommendations have been made for understanding the heavy air pollution in Beijing:

(1)Compared to the increasing ozone during APEC period, average ozone concentration decreased significantly in the downwind areas of Beijing due to larger ratios of $VOCs/NO_x$.

In order to decrease the ozone concentration in Beijing, emissions of VOCs should be reduced larger than those of $NO_x$ in Beijing. The collaborative control of various pollutants is becoming very important in Beijing.

(2)As air pollution is a regional problem, therefore, the simultaneous implementation of a regional prevention and control mechanism is necessary to promote continuous air quality improvement in Beijing.

(3)Many of the world's thriving cities are struggling with serious air pollution, Beijing's experience in controlling ozone against a backdrop of rapid expansion during air quality assurance periods is a story that should be shared with other emerging economies and burgeoning cities.

**Acknowledgments**

This study was supported by the Commonwealth Project of the Ministry of Environmental Protection (NO.201409005) and the National Key Technology R&D Program (2014BAC23B03). For detailed data,please see website http://zx.bjmemc.com.cn/ or send an email to15001195306@163.com.

The English in this document has been checked by at least two professional editors, both native speakers of English.

Liu,H.R.,Liu,C.,Xie,Z.Q.,Li,Y.,Huang,X.,Wang,S.S.,Xu,J.,and Xie,P.H.:A paradox for air pollution controlling in China revealed by"APEC Blue" and "Parade Blue",Scientific Reports,6:34408.DOI: 10.1038/srep34408.

[revised manuscript text omitted]

---

## Author Comment (AC2) · 1 Jan 2017

This manuscript has been further edited for language by Essaystar (http://essaystar.com/Service.html), acompany dedicated to helping international researchers publish their findings in the best English language journals possible. The study was restricted by some necessary conditions such as: we lack of observed data for VOCs during 2008–2015, NOx during 2004–2007, meteorological elements during 2004–2009. So the we cannot analyzed them perfectly. In order to make deep analysis and find valuable scientific information on the O3 pollution in Beijing,we have done the following aspects of work 1. We add some further analysis and modify the reviewer's opinion. We applied the NO2 and HCHO VCDs from the OMI products in urban Beijing

(GY site, 116.33° E, 39.93° N) and combined the corresponding ratios to analyze the chemical sensitivity of PO3. 2. In order to explain the regional reductions effect ,we increased analysis and changed some orders of the text. 3. We further polish and improve the English language in this article and the proof is as follows: If there some more revisions,we will further modify. Thank you! The annex is the english polish proof.

Please also note the supplement to this comment:
http://www.atmos-chem-phys-discuss.net/acp-2016-508/acp-2016-508-AC2-supplement.pdf

[Figure]

English polish proof.

**Review Certificate**
* * *
To whom it may concern:

This memo certifies that one of our clients has contracted our academic editing service for the following file.

Order Number:
**P-201604200528izp**
Word Count:
**3388 words**
Date of the review:
**04/20/16** ( MM/DD/YY )

The English review was conducted using a two-stage process, in which a junior editor first reviewed the file, and then a senior editor conducted a final and more thorough review. All of our editors are native English-speaking professionals.

Documents receiving this certification should be English-ready for publication; however, the author has the ability to accept or reject our suggestions and changes.

We would like to emphasize that our service targets grammar and language edits. We do not rewrite the documents from scratch. If you are dissatisfied with specific revisions, please contact service@essaystar.com.

Essaystar Group

+1-208-975-4235
EssayStar, 93 S Jackson St, Seattle, WA 98104

**Fig. 1.** English polish proof